# Domain Restriction via Multi SAE Layer Transitions

Elias Shaheen[1]    Avi Mendelson[1]

## Abstract

The general-purpose nature of Large Language Models (LLMs) presents a significant challenge for domain-specific applications, often leading to out-of-domain (OOD) interactions that undermine the provider's intent. Existing methods for detecting such scenarios treat the LLM as an uninterpretable black box and overlook the internal processing of inputs. In this work we show that layer transitions provide a promising avenue for extracting domain-specific signature. Specifically, we present several lightweight ways of learning on internal dynamics encoded using a sparse autoencoder (SAE) that exhibit great capability in distinguishing OOD texts. Building on top of SAEs representation transitions enables us to better interpret the LLM internal evolution of input processing and shed light on its decisions. We provide a comprehensive analysis of the method and benchmark it with the gemma-2 2B and 9B models. Our results emphasize the efficacy of the internal process in capturing fine-grained input-related details.[1]

## 1. Introduction

Large language models are increasingly used as interfaces for information access and automation in real-world systems (Chen et al., 2023; Hu et al., 2024). In many deployments, LLMs are embedded in agentic workflows that plan, call tools, and coordinate specialized components to accomplish user tasks.(Yao et al., 2023; Schick et al., 2023) To improve reliability and cost-efficiency, service providers often restrict these systems to a specific domain and deploy domain-specialized agents rather than a single general-purpose assistant(Chen et al., 2023; Hu et al., 2024). As

[1]Technion – Israel Institute of Technology, Haifa, Israel. Correspondence to: Elias Shaheen <eliasshaheen@campus.technion.ac.il>, Avi Mendelson <mendlson@technion.ac.il>.

*Proceedings of the 43${}^{rd}$ International Conference on Machine Learning*, Seoul, South Korea. PMLR 306, 2026. Copyright 2026 by the author(s).

[1]Code will be released in the future.

a result, ensuring that requests are handled by the appropriate agent has become a key requirement for practical LLM-based systems.(Hu et al., 2024)

However, these systems introduce additional complexity: multiple components must coordinate through language and intermediate decisions.(Liu et al., 2025; Zhou et al., 2024; Yao et al., 2023) Since LLM outputs can be unreliable and sometimes hallucinated, errors can cascade across steps and cause the workflow to diverge from the intended goal.(Huang et al., 2025; Liu et al., 2024) This results in failed task execution, poor user experience, and wasted compute and operational overhead for deployers.

This motivates a gating mechanism that assesses whether an input request is appropriate for a given agent(Hu et al., 2024; Chen et al., 2023; Shazeer et al., 2017). Formally, given a request $x$ and an agent scope $S$, the goal is to decide whether $x$ is in-scope (to be handled by the agent) or out-of-scope (to be rejected or rerouted). We view this as an out-of-distribution (OOD) detection problem in language, where in-scope requests follow the target distribution induced by $S$. Performance is evaluated using standard OOD discrimination metrics.(Hendrycks & Gimpel, 2018; Liang et al., 2020)

Prior work has studied OOD detection for neural networks and language models by leveraging different signals available in transformer architectures.(Podolskiy et al., 2022; Sun et al., 2022) A common post-hoc baseline uses output confidence, e.g., maximum softmax probability (MSP) (Hendrycks & Gimpel, 2018). Beyond output scores, many approaches exploit internal representations, using feature-space distances or fitted density models to separate in-distribution inputs from OOD samples (Lee et al., 2018; Podolskiy et al., 2022; Sun et al., 2022). In the NLP setting, pretrained transformers have also been shown to substantially improve robustness to distribution shift and out-of-domain inputs (Hendrycks et al., 2020; Uppaal et al., 2023). For broader context and additional families of methods, we refer to recent surveys (Lu et al., 2025; Lang et al., 2023).

While these methods can be effective, they often offer limited transparency into *why* an input is judged as out-of-scope (Räuker et al., 2023; Elhage et al., 2022), and practical constraints may limit access to labeled OOD data or the ability to fine-tune models.(Lu et al., 2025) This motivates

approaches that are both interpretable and lightweight to deploy, while maintaining strong OOD discrimination in realistic settings.

In this work, we investigate whether transformer internals provide a reliable signal for deciding in-scope versus out-of-scope requests. Concretely, we analyze the evolution of residual-stream representations and map them into an interpretable feature space using sparse autoencoders (SAEs).(Cunningham et al., 2023) Using only in-scope examples, we characterize a reference signature of the model's internal behavior and use deviations from this signature to score inputs at test time. This design is lightweight and does not require labeled OOD data or fine-tuning of the base model. We evaluate the approach under both near- and far-OOD settings and provide analysis linking detection performance to internal representation dynamics.

Our contributions are:

- We propose an *ID-only* scope-gating method that detects out-of-scope text by modeling *depthwise transitions* of sparse, interpretable SAE features, without fine-tuning the base LLM or using labeled OOD data.

- We introduce an SAE→SDR representation for transformer-layer trajectories and a modular sequential scoring framework, and we compare lightweight backends (first-order Markov, Hierarchical Temporal Memory (HTM), and an RNN predictor) for anomaly scoring.

- We provide representation analyses on Gemma2-2B characterizing where domain-consistent structure emerges across depth and how transition-based modeling differs from static overlap.

- We evaluate the resulting gate on both near- and far-OOD benchmarks, and we provide interpretability analyses (including a hard-OOD boundary case study) linking detection behavior to decoded SAE feature transitions.

## 2. Related Work and Background

### 2.1. Sparse Autoencoders

**Motivation: sparse feature dictionaries for superposed representations.** Transformer residual streams are high-dimensional and distributed: many distinct latent factors can be represented in overlapping directions, a phenomenon often described as *superposition*. This makes single-neuron interpretations unreliable and motivates learning an explicit *feature dictionary* that decomposes activations into a sparse set of (approximately) reusable components (Elhage et al., 2022).

**SAE formulation.** Given a layer $\ell$ and a residual-stream activation $h_{\ell,t} \in \mathbb{R}^d$ at token position $t$, a sparse autoencoder (SAE) learns an encoder–decoder pair $(\mathrm{Enc}_\ell, \mathrm{Dec}_\ell)$ such that

$$z_{\ell,t} = \mathrm{Enc}_\ell(h_{\ell,t}) \in \mathbb{R}^D, \qquad \hat{h}_{\ell,t} = \mathrm{Dec}_\ell(z_{\ell,t}), \quad (1)$$

with $D \gg d$ and $z_{\ell,t}$ encouraged to be sparse. Training typically minimizes a reconstruction loss plus a sparsity penalty, e.g.,

$$\min_{\mathrm{Enc}_\ell, \mathrm{Dec}_\ell} \mathbb{E}\big[\|h_{\ell,t} - \hat{h}_{\ell,t}\|_2^2\big] + \lambda\|z_{\ell,t}\|_1, \quad (2)$$

or related sparsity constraints. Intuitively, $\mathrm{Dec}_\ell$ learns a set of dictionary atoms (columns) and the sparse code $z_{\ell,t}$ selects a small subset of them to explain $h_{\ell,t}$.

**Interpretability and empirical evidence.** Recent work shows that SAEs trained on LLM activations can yield *highly interpretable* features, often aligning with human-recognizable concepts, syntax, or task-related patterns, and improving over direct neuron-level inspection (Cunningham et al., 2023; Paulo et al., 2025). This line of work operationalizes the "features not neurons" view of mechanistic interpretability by replacing polysemantic neuron activations with sparse, compositional feature activations.

**Why SAEs are useful for our setting.** Our goal is to detect *domain restriction* signals from the *internal dynamics* of an LLM using only in-domain data. SAEs provide two properties that are particularly useful here: (i) a *common coordinate system* across examples (feature indices) that makes trajectories comparable across inputs and layers (Marks et al., 2025; Arad et al., 2025); and (ii) a *sparsified* representation where Top-$k$ active features can be treated as a discrete state suitable for transition modeling.(Cui et al., 2016) This representation is suitable to lightweight sequential models (e.g., Markov/HTM/RNN backends) than dense residual vectors, and it supports post-hoc analysis by mapping active indices back to feature descriptions.

**Related directions and tooling.** Several closely related efforts learn and analyze sparse feature dictionaries in transformers, including variations on dictionary learning and cross-layer feature mappings (e.g., *crosscoders*) (Lindsey et al., 2024). On the tooling side, open-source libraries such as `SAELens` facilitate training and applying SAEs to modern LLMs and standardizing evaluation/visualization workflows (Bloom et al., 2024), additionally we use Neuronpedia for leveraging features data such as text label and global densities. (Lin, 2023) In our work, we treat SAEs as a fixed, pretrained decomposition of each layer's residual stream and focus on what can be learned *on top* of these features from in-domain data alone—namely, whether *depthwise transition regularities* form a stable, domain-specific signature suitable for scope gating.

## 2.2. OOD Detection

Deployed models inevitably get exposed to out-of-distribution samples, as opposed to the closed-world assumption (Vapnik, 1991), such scenario might cause model's to behave in ways that are not expected and even sometimes harmful, this motivates the detection of such outliers. Traditionally OOD detection was the paradigm of detecting test-time samples that were not present in the training data of the model (Sun et al., 2022; Lee et al., 2018; Hendrycks & Gimpel, 2018), however and since LLMs are trained on extremely broad and heterogeneous web-scale corpora, the boundary between ID and OOD became blurry.(Lang et al., 2023) The feild of OOD detection splits into different settings based on the presence of OOD samples and labels for the training samples (Lang et al., 2023), the method this paper discusses operates in the absence of both. Naturally the field migrated into exploring detecting OOD in the inputs to Large pre-trained encoders like BERT(Hendrycks et al., 2020; Podolskiy et al., 2022; Chen et al., 2022; Devlin et al., 2019; Uppaal et al., 2023), we explore a similar paradigm in their counterparts, the decoder-only Large Pretrained Models.

Closest to our experimental setting is (Zhang et al., 2025), with the difference of them using a fine-tune variant of the model to calculate the anomaly score, whereas we adapt an auxlary lightweight model on the internal activation transitions to do so.

## 3. Methodology

Our method, illustrated in Figure 1, is a four-stage pipeline designed to detect out-of-domain inputs by analyzing the internal activation flow of an LLM.

### 3.1. Activation Extraction

Given an input sequence of tokens $x = (x_1, \ldots, x_T)$, we extract post layer residual-stream hidden states across a set of $L$ continuous transformer layers and apply a mask to exclude padding positions in all subsequent steps.[2] Let $h_{\ell,t} \in \mathbb{R}^d$ denote the hidden state at layer $\ell \in \{1, \ldots, L\}$ and token position $t \in \{1, \ldots, T\}$. This yields a layerwise collection of token activations $\{h_{\ell,1}, \ldots, h_{\ell,T}\}_{\ell=1}^L$.

### 3.2. Sparse Feature Decomposition with SAEs

To obtain an interpretable, sparse representation, we pass each token activation $h_{\ell,t}$ through a pre-trained layer-specific sparse autoencoder (SAE) encoder:

$$z_{\ell,t} = \text{Enc}_\ell(h_{\ell,t}) \in \mathbb{R}^D, \qquad (3)$$

---

[2]We also evaluated using only the final token representation; token-aggregated representations provided a more stable signal in our SAE setting

where $D \gg d$. The resulting vectors $z_{\ell,t}$ are sparse, and their non-zero entries correspond to learned features that are approximately mono-semantic.

**Token-axis pooling.** For each layer $\ell$, we aggregate token-level SAE features into a single layer representation by averaging over token positions:

$$\bar{z}_\ell = \frac{1}{\sum_{t=1}^T m_t} \sum_{t=1}^T m_t \, z_{\ell,t}, \qquad (4)$$

where $m_t \in \{0, 1\}$ is the *padding mask*, 1 denoting non-padding token. This produces a pooled sparse feature trajectory $\{\bar{z}_1, \ldots, \bar{z}_L\}$.

**Global-density feature masking.** Some SAE features are overly frequent and carry little domain-specific information. We therefore prune features using global activation density statistics (e.g., Neuronpedia-reported densities). Let $\rho_{\ell,j}$ denote the global activation density of feature $j$ at layer $\ell$. We construct a binary mask $g_\ell \in \{0, 1\}^D$ that removes features above a density threshold $\theta$:

$$g_{\ell,j} = \mathbb{1}\left[\rho_{\ell,j} \leq \theta\right], \qquad \tilde{z}_\ell = \bar{z}_\ell \odot g_\ell, \qquad (5)$$

where $\odot$ is element-wise multiplication.

**Top-$k$ binarization.** Finally, we convert $\tilde{z}_\ell$ into a binary sparse distributed representation (SDR) by selecting the indices of the top-$k$ remaining features by magnitude:

$$A_\ell = \text{TopK}(\tilde{z}_\ell, k), \qquad s_\ell[j] = \mathbb{1}[j \in A_\ell], \qquad (6)$$

yielding a binary sequence $\{s_1, \ldots, s_L\}$ used by downstream sequential scoring modules.

### 3.3. Sequential Anomaly Scoring

For an input $x$ we obtain a depthwise SDR sequence $\{s_\ell\}_{\ell=1}^L$, where $s_\ell \in \{0, 1\}^D$ has exactly $k$ active bits. Let $A_\ell(x) = \{j \in [D] \mid s_\ell[j] = 1\}$ be the active set at layer $\ell$. Each backend defines a per-layer anomaly $a_\ell(x)$ for $\ell = 2, \ldots, L$ and we aggregate

$$S(x) = \frac{1}{L-1} \sum_{\ell=2}^L a_\ell(x). \qquad (7)$$

**Markov (default).** From ID data we count adjacent-layer co-activations $C_\ell(i, j) = \#\{x \in \mathcal{D}_{ID} : i \in A_{\ell-1}(x), j \in A_\ell(x)\}$ and marginals $N_\ell(i) = \sum_j C_\ell(i, j)$. With Laplace smoothing $\alpha$, we set

$$p_\ell(j \mid i) = \frac{C_\ell(i, j) + \alpha}{N_\ell(i) + \alpha D}. \qquad (8)$$

At test time we score transitions by the negative mean log-likelihood over active pairs:

$$a_\ell(x) = -\frac{1}{|A_{\ell-1}||A_\ell|} \sum_{i \in A_{\ell-1}} \sum_{j \in A_\ell} \log p_\ell(j \mid i), \qquad (9)$$

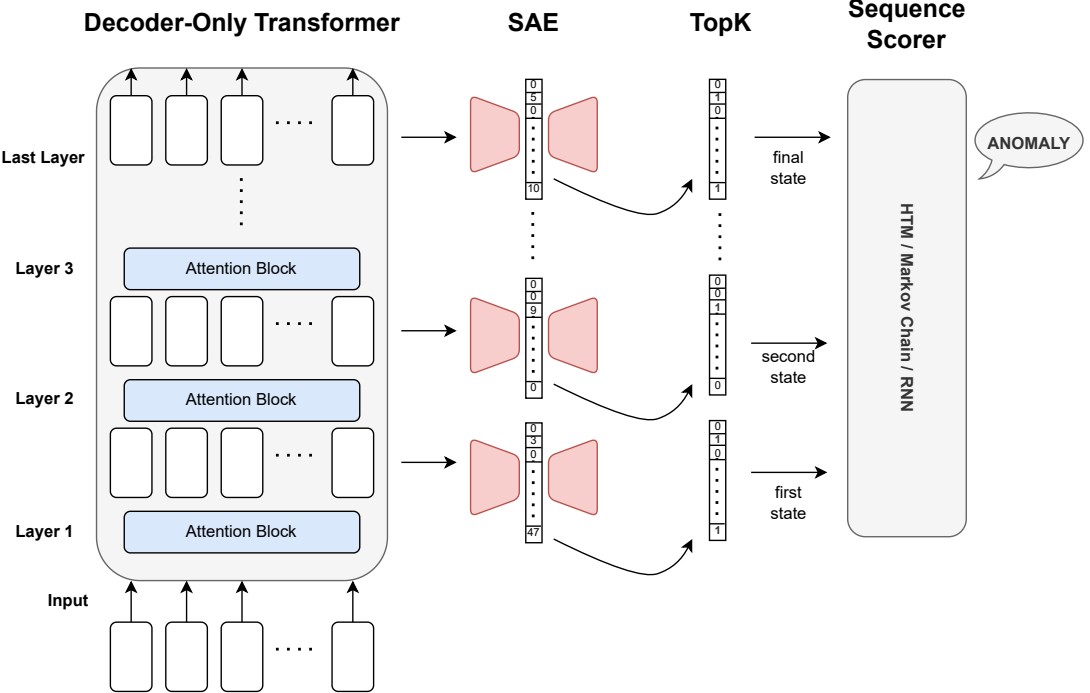

*Figure 1.* Pipeline overview. Given an input, we extract residual-stream activations across layers, encode them with layer-wise SAEs, pool over tokens, mask globally frequent features, and binarize via Top-$k$ to obtain an SAE→SDR depth trajectory. A lightweight sequential scorer (Markov/HTM/RNN) trained on ID-only data assigns an anomaly score from depthwise transitions.

assigning unseen transitions the smoothed floor probability.

**HTM and RNN.** As higher-order baselines, we also score sequences using (i) HTM temporal memory anomaly (fraction of active bits not predicted), and (ii) an LSTM/GRU next-layer predictor trained on ID sequences with per-bit BCE loss; full details are in the appendix.

## 4. Experiments

### 4.1. Experimental Setup

We run all experiments on pretrained **Gemma2-2B** and **Gemma2-9B**.(Team et al., 2024) From selected transformer blocks we extract residual-stream activations over the first 512 tokens and encode them using a pretrained **16k-dimensional sparse autoencoder (SAE)**: layers 16–24 for Gemma2-2B and layers 38–41 for Gemma2-9B. SAE activations are mean-pooled over tokens and converted to a binary sparse distributed representation (SDR) via Top-$k$ selection. Unless stated otherwise, we use $k=10$. We evaluate ID-only OOD detection under two regimes: **far-OOD**, where ID is 20 Newsgroups and OOD is drawn from SST-2, MNLI, RTE, IMDB, and CLINC150; and **near-OOD**, where ID and OOD share the same dataset but differ by semantic class, using AGNews (4 one-vs-all splits), ROSTD, SNIPS, and CLINC150 (see Table 3 for split details). Our detector operates on depth-wise SAE-SDR sequences using

three alternative sequential backends: a sparse first-order **Markov** transition model, **HTM** (Cui et al., 2016), and an **RNN** (LSTM) trained to predict next-layer activations with BCE; unless stated otherwise we use Markov. As a baseline we use the **likelihood-ratio (LR)** criterion between a base model and an in-domain fine-tuned model as it achieves SOTA (Zhang et al., 2025). We report AUROC, AUPR (OOD as positive), and FPR@95%TPR, averaged over three seeds where applicable.

### 4.2. Representation Analysis

#### 4.2.1. DOMAIN CHARACTERIZATION

We begin by characterizing where and how strongly a domain signal appears in the SAE feature space across depth. For an input $x$, recall that our representation produces a masked pooled vector $\tilde{z}_\ell(x)$ at each layer $\ell$, and an active feature set

$$A_\ell(x) = \text{TopK}(\tilde{z}_\ell(x), k). \quad (10)$$

For a fixed $k$, we measure *within-domain consistency* at each layer by sampling pairs of in-domain inputs $(x, x')$ and computing the Jaccard similarity

$$J_\ell(x, x') = \frac{|A_\ell(x) \cap A_\ell(x')|}{|A_\ell(x) \cup A_\ell(x')|}. \quad (11)$$

We report the mean of $J_\ell$ over sampled pairs as a function of layer and sweep $k$, shown in Figure 2.

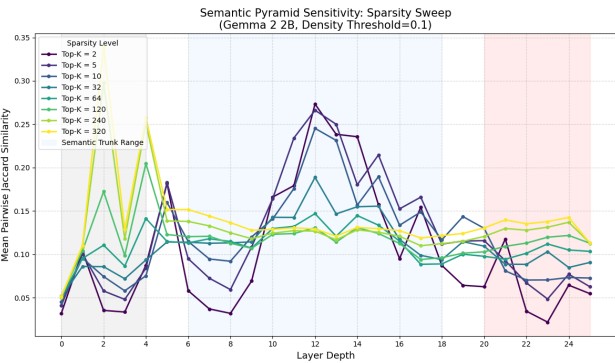

*Figure 2.* Mean Jaccard similarity across layers for different Top-$k$ settings (computed on all pairs of a subset of 1000 in-domain samples).

Across $k$, we observe a characteristic depth profile: early layers exhibit lower agreement in top activating features, mid-layers show increased similarity in top activating features, and late layers become more input-specific. This motivates selecting a moderate $k$ that preserves stable shared structure while avoiding excessive noise (we use $k{=}10$ in subsequent analyses unless stated otherwise). The peak around mid-layers suggests that domain-consistent features are most prominent in this region in most activating features, which motivates examining whether *transitions* across layers carry additional domain structure.

### 4.2.2. DYNAMIC ANALYSIS

The static agreement analysis indicates that in-domain inputs share features in particular depth regions, but it does not reveal whether these features are domain specifics. We therefore analyze the specificity of the features across layers and transitions between them.

Given a domain $D$, for each start layer $\ell$ and hop length $N$, we construct a registry of all $N$-hop feature tuples observed in the in-domain data:

$$\mathcal{V}_{\ell,N} := \left\{ (u_0, \dots, u_N) \;\middle|\; \begin{array}{l} \exists x \in D, \\ \forall i = 0, \dots, N : \; u_i \in A_{\ell+i}(x) \end{array} \right\}.$$
$$(12)$$

We consider $N \in \{0, 1, 2\}$, where $N = 0$ corresponds to a static registry.

**Trajectory-based scoring.** For a sample $x$, define the set of induced trajectories starting at layer $l$ with hop length $H$ as

$$\mathcal{T}_{l,H}(x) = A_l(x) \times A_{l+1}(x) \times \cdots \times A_{l+H}(x), \quad (13)$$

with $|\mathcal{T}_{l,H}(x)| = \prod_{i=0}^{H} |A_{l+i}(x)|$. We score how *typical* these trajectories are under the in-domain registry by the fraction of induced tuples present in $\mathcal{T}_{l,H}(x)$:

$$\text{Score}_H(x, l) = \frac{|\mathcal{T}_{l,H}(x) \cap \mathcal{V}_{l,H}|}{|\mathcal{V}_{l,H}|}. \quad (14)$$

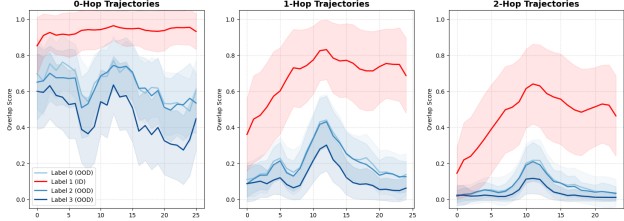

*Figure 3.* Layer-wise trajectory validity scores for hop lengths $H \in \{0, 1, 2\}$ (Top-$k$=10). Lines show mean and shaded regions indicate $\pm 1$ standard deviation.

Figure 3 shows that (i) different semantic classes induce distinguishable depthwise trajectories, and (ii) longer hop lengths exhibit increased variance within a class, consistent with encoding more context-specific information. However, the registry-based construction is inherently coarse: it accumulates many incidental tuples, which increases noise and can lead to elevated FPR@95 as can be seen in 1. This motivates replacing the explicit registry with learned sequential scoring functions that emphasize recurring transition structure while down-weighting spurious co-activations.

*Table 1.* Mean OOD Detection Metrics across Trajectory Hop Lengths

| Hop | AUROC | AUPR | FPR@95 | Pairs Evaluated |
|-----|-------|------|--------|-----------------|
| 0 | 0.8445 | 0.8092 | 0.4451 | 12 |
| 1 | 0.9159 | 0.9226 | 0.3902 | 12 |
| 2 | 0.9128 | 0.9204 | 0.4054 | 12 |

**Takeaway.** The static overlap results indicate that in-domain inputs share stable high activating feature structure in mid-depth layers, while the trajectory analysis shows that depthwise transitions carry additional domain-specific regularities. However, explicit tuple registries are coarse and accumulate incidental patterns, motivating a learned sequential scoring function that can emphasize recurring transition structure while suppressing spurious co-activations. In the ablation section, we evaluate sequential scoring backends trained only on in-scope data.

Unless stated otherwise, we report results using the first-order Markov backend; alternative backends (HTM and RNN) are compared in the ablation study.

### 4.3. Main Results

We evaluate scope gating as OOD detection under two regimes: *far-OOD*, where out-of-scope data differs in both semantics and domain/covariates, and *near-OOD*, where in-scope and out-of-scope inputs share similar style but differ semantically. We report AUROC, AUPR (OOD as positive), and FPR@95%TPR.

### 4.3.1. BASELINES

Our primary comparator is the likelihood-ratio (LR) criterion between a pretrained base LLM and an in-scope-adapted variant, which has been shown to be a strong OOD detector in an ID-only setting. (Zhang et al., 2025) This baseline is particularly relevant in our setting because it does not require labeled or OOD data and naturally fits the scope-gating interpretation of OOD detection.

### 4.3.2. FAR-OOD RESULTS

In the far-OOD setting, the in-scope dataset 20NG is fixed and multiple out-of-scope datasets are used as OOD sources. This evaluates coarse rejection under substantial domain/covariate shift. Results are reported in Table 2.

### 4.3.3. NEAR-OOD RESULTS

In the near-OOD setting, in-scope and out-of-scope examples are drawn from the same dataset split by semantic classes, preserving similar surface form while changing intent/topic. This setting stresses fine-grained semantic discrimination, and is commonly used to evaluate scope detection.Results are reported in Table 3.

**Discussion.** Near-OOD detection is substantially more challenging than far-OOD, as in- and out-of-scope examples share surface form and differ only in fine-grained semantics. On SNIPS, ROSTD and AGNews our method is competitive with the LR baseline and achieves comparable FPR95, indicating that transition regularities in internal representations can capture intent-level shifts when the underlying domains are semantically coherent and sufficiently coarse.

On CLINC150, however, performance degrades sharply for the 2B model, most notably in AUPR. We attribute this failure to a *representation resolution mismatch* rather than a modeling artifact. CLINC150 defines 150 highly specific intents spanning very different tasks (e.g., mechanical instructions, banking queries, authentication issues, customer support, etc.), while our method operates on a finite set of SAE features whose semantic granularity is necessarily coarser.

As a result, many distinct intents collapse onto overlapping subsets of SAE features and therefore induce highly similar, high-entropy trajectories across depth touching upon many abstract concepts. In this regime, different intents are not separated by stable, low-entropy manifolds in representation space, but instead share large portions of their activation patterns. Since the sequential scorer relies on consistent inter-layer transition structure to distinguish in-domain from out-of-domain behavior, this representational aliasing makes reliable discrimination impossible. The stronger performance of the 9B model suggests that increased representational capacity partially alleviates this bottleneck, but

does not remove it entirely. We therefore view CLINC150 as illustrating a fundamental limitation of representation-dynamics based domain restriction: when the task taxonomy is significantly finer-grained than the resolution of the underlying features, domain boundaries cease to be well-defined in internal representation space.

## 4.4. Case Study: Symmetric "Hard-OOD" Boundary Examples in AGNews

Near-OOD settings can contain genuine semantic overlap between the ID and OOD classes, making some OOD inputs effectively *in-domain in intent*. To illustrate what our detector learns beyond dataset labels, we conduct a focused case study on AGNews labels **Business (2)** and **Sci/Tech (3)** using the Markov backend from Section 3.3. For each direction, we train the Markov transition model on ID only and then identify *hard OOD* examples: OOD test inputs with the *lowest* anomaly score $S(x)$ (i.e., those most confidently judged as ID by the transition model). Concretely, hard OOD corresponds to false negatives under the OOD-as-positive convention.

**Observation: hard OOD is semantically "in-domain" in both directions.** When training on **Business** and testing on **Sci/Tech** as OOD, hard-OOD Sci/Tech articles are predominantly *finance/earnings/market* stories (often about tech companies), and thus resemble Business in content. Conversely, when training on **Sci/Tech** and treating **Business** as OOD, hard-OOD Business articles frequently describe *telecom infrastructure, software releases, and product launches*, resembling Sci/Tech. This symmetry indicates that many apparent "errors" arise from *true boundary overlap* rather than arbitrary model failure.

**SAE-transition evidence: hard OOD follows ID-like internal dynamics.** Beyond the raw prompt text, we inspect which SAE feature transitions explain the low anomaly. For each hard-OOD example, we extract the top contributing active feature-pairs $(i \rightarrow j)$ across depth (highest $-\log p_\ell(j \mid i)$ among active pairs; see Section 3.3). We decode feature indices to human-readable labels via Neuronpedia.

The resulting explanations show that hard OOD tends to activate and transition through features whose semantics match the ID domain signature (e.g., market/earnings concepts under Business-ID; network/software concepts under Sci/Tech-ID), supporting the hypothesis that the Markov score tracks *domain-consistent internal trajectories* rather than superficial cues.

*Table 2.* Far-OOD results (ID-only training).

| ID | OOD | Method | AUROC ↑ | AUPR (OOD) ↑ | FPR95 ↓ |
|----|-----|--------|---------|--------------|---------|
| 20NG | SST-2 | Gemma2-2b + 16k_SAE + Markov | 0.9782 | 0.7683 | 0.1101 |
| | | Gemma2-9b + 16k_SAE + Markov | 0.9866 | 0.8713 | 0.0665 |
| | | Gemma2-2b LR baseline | **0.9988** | **0.9871** | **0.0027** |
| | | Gemma2-9b LR baseline | 0.9980 | 0.9994 | 0.004 |
| 20NG | MNLI | Gemma2-2b + 16k_SAE + Markov | 0.9699 | 0.9680 | 0.1984 |
| | | Gemma2-9b + 16k_SAE + Markov | 0.9763 | 0.9768 | 0.1628 |
| | | Gemma2-2b LR baseline | **0.9989** | **0.9988** | **0.0027** |
| | | Gemma2-9b LR baseline | 0.9982 | 0.997 | 0.003 |
| 20NG | RTE | Gemma2-2b + 16k_SAE + Markov | 0.9850 | 0.6422 | 0.0325 |
| | | Gemma2-9b + 16k_SAE + Markov | 0.9884 | 0.7333 | 0.0469 |
| | | Gemma2-2b LR baseline | **0.9991** | 0.9950 | **0.0022** |
| | | Gemma2-9b LR baseline | 0.9987 | **0.9994** | 0.003 |
| 20NG | IMDB | Gemma2-2b + 16k_SAE + Markov | 0.9171 | 0.9058 | 0.6243 |
| | | Gemma2-9b + 16k_SAE + Markov | 0.9540 | 0.9544 | 0.3361 |
| | | Gemma2-2b LR baseline | **0.9978** | **0.9992** | **0.0072** |
| | | Gemma2-9b LR baseline | 0.9945 | 0.9840 | 0.017 |

**Business-ID (2) → hard OOD from Sci/Tech (3)**
*"Gateway Reports Smaller Quarterly Loss . . . continues to restructure . . . integrate its acquisition . . . "*
**Decoded transition evidence:**

- 16→17: *terms related to business acquisitions and financial transactions → references to significant events and metrics in financial contexts*
- 18→19: *terms related to acquisitions and financial transactions → references to financial data and reporting metrics*

**Sci/Tech-ID (3) → hard OOD from Business (2)**
*"Cingular to upgrade wireless data network . . . handle high-speed data transmissions . . . "*
**Decoded transition evidence:**

- 17→18: *references to financial institutions and related operational terms → references to "data" and its contexts in research and analysis*

**Takeaway.** Across both directions, the *lowest-anomaly OOD* examples are precisely those whose *content* overlaps the ID domain (e.g., earnings-driven tech reporting or telecom/software business news), and their *internal SAE trajectories* expose corresponding domain-aligned feature transitions. This supports the interpretation that our method performs *semantic scope gating* by modeling consistent depthwise feature dynamics, and that many "mistakes" are explainable boundary cases rather than spurious failures. Additional examples are provided in Section D.

## 4.5. Runtime and Practical Overhead

Our method does not apply SAEs to all transformer layers. In the main setting, we extract SAE features from a contiguous block of deeper layers, using up to the first 512

tokens per input. The dominant computational cost is SAE feature extraction; fitting and scoring the transition model are comparatively small.

*Table 4.* Runtime breakdown for a representative Gemma-2-2B setup using a 16k SAE and layers 16–24. The benchmark uses 100 ID training samples, 50 ID test samples, and 50 OOD test samples, each with at least 512 raw tokens.

| Stage | Time (s) |
|-------|----------|
| Train extraction | 57.43 |
| Train fit | 1.58 |
| ID extraction | 29.27 |
| ID scoring | 0.0945 |
| OOD extraction | 25.45 |
| OOD scoring + metrics | 0.0990 |
| Full pipeline total | 151.92 |

The results show that feature extraction dominates runtime, while Markov fitting and scoring add negligible overhead once SAE features have been extracted. Thus, the method is more expensive than pure output-based scoring, but avoids fine-tuning and provides an interpretable white-box signal from internal feature dynamics.

## 4.6. Ablations

### 4.6.1. SEQUENTIAL BACKEND ABLATION

**Setup.** We compare three sequential backends operating on the same SAE-derived binary SDR sequences across transformer depth: (i) **HTM**, a high-order temporal memory

*Table 3.* Near-OOD results (ID-only training). Splits: **AGNews** uses 4 one-vs-all runs (reported as $c_3/c_2/c_1/c_0$ in each metric cell). **ROSTD** uses the dataset's predefined in-domain classes (13) vs. its OOD set. **SNIPS** uses ID intents {AddToPlaylist, PlayMusic, RateBook, SearchCreativeWork, SearchScreeningEvent} and OOD intents {GetWeather, BookRestaurant}. **CLINC150** uses the dataset's predefined in-domain classes (150) vs. its OOD set.

| Dataset | Method | AUROC ↑ | AUPR (OOD) ↑ | FPR95 ↓ |
|---|---|---|---|---|
| AGNews | Gemma2-2b + 16k_SAE + Markov | **0.90** /0.98 /0.92 /0.88 | **0.96** /0.99 /0.97 /0.95 | 0.50 /**0.03** /**0.41** /**0.51** |
| | Gemma2-9b + 16k_SAE + Markov | 0.87 /0.90 /0.83 /0.81 | 0.95 /0.96 /0.93 /0.92 | **0.45** /0.31 /0.54 /0.59 |
| | Gemma2-2b LR baseline | 0.75 /0.76 /0.82 /0.73 | 0.89 /0.90 /0.93 /0.88 | 0.76 /0.68 /0.67 /0.80 |
| | Gemma2-9b LR baseline | 0.73 /0.74 /0.79 /0.75 | 0.89 /0.86 /0.90 /0.87 | 0.79 /0.77 /0.70 /0.75 |
| ROSTD | Gemma2-2b + 16k_SAE + Markov | 0.9532 | **0.9548** | **0.1757** |
| | Gemma2-9b + 16k_SAE + Markov | 0.9445 | 0.9454 | 0.2068 |
| | Gemma2-2b LR baseline | **0.9726** | 0.9455 | 0.1690 |
| | Gemma2-9b LR baseline | 0.9549 | 0.9165 | 0.2597 |
| SNIPS | Gemma2-2b + 16k_SAE + Markov | 0.9699 | 0.9338 | 0.1525 |
| | Gemma2-9b + 16k_SAE + Markov | 0.9001 | 0.8287 | 0.3150 |
| | Gemma2-2b LR baseline | 0.9746 | 0.9423 | 0.1528 |
| | Gemma2-9b LR baseline | **0.9798** | **0.9546** | **0.0873** |
| CLINC150 | Gemma2-2b + 16k_SAE + Markov | 0.7387 | 0.0127* | 0.8333 |
| | Gemma2-9b + 16k_SAE + Markov | 0.8364 | **0.7278** | **0.4825** |
| | Gemma2-2b LR baseline | **0.8661** | 0.6449 | 0.5607 |
| | Gemma2-9b LR baseline | 0.8494 | 0.6296 | 0.6440 |

model; (ii) **RNN**, an LSTM next-layer predictor trained with a BCE objective; and (iii) **Markov**, a first-order transition model estimating sparse adjacent-layer transition probabilities. All models are trained on in-distribution (ID) data only and evaluated for OOD detection using AUROC, AUPR, and FPR@95 (lower is better).

**Aggregated results.** To summarize the four ID-class runs, we report the mean performance across runs in Table 5.

*Table 5.* Mean OOD detection performance across four ID-class runs for different sequential backends.

| Method | AUROC ↑ | AUPR ↑ | FPR@95 ↓ |
|---|---|---|---|
| HTM | 0.912 | 0.963 | 0.384 |
| Markov | **0.919** | **0.968** | **0.371** |
| RNN | 0.910 | 0.965 | 0.392 |

**Interpretation.** A simple first-order Markov model matches or slightly outperforms both high-order backends (HTM and RNN) on average. This indicates that, under the current SAE-SDR representation, **most of the discriminative signal for OOD detection is already present in adjacent-layer transition statistics**. Higher-order sequence modeling provides no consistent advantage, suggesting that transformer depth dynamics in this setting are close to first-order Markovian for the purpose of ID/OOD separation.

### 4.6.2. STATIC SAE BASELINE: IS TRANSITION MODELING NECESSARY?

To isolate the contribution of transition modeling from the contribution of the SAE representation itself, we compare against a static multi-layer SAE baseline. This baseline uses the same Gemma-2-2B 16k SAE representation and the same layer range as our Markov model, but removes the transition component. For each layer, we fit a Gaussian distribution to the pooled SAE activations of the in-domain training set and score test examples using the Mahalanobis distance to the in-domain distribution. We then average scores across layers.

*Table 6.* Static SAE baseline versus transition modeling. Results report AUROC. for AGNews we report the average of the 4 classes

| Method | AGNews | SNIPS | CLINC150 |
|---|---|---|---|
| LR baseline | 0.787 | **0.975** | **0.866** |
| Static Mahalanobis | 0.905 | 0.862 | 0.791 |
| Markov (ours) | **0.915** | 0.960 | 0.830 |

The static SAE baseline is already strong, confirming that multi-layer SAE features contain useful domain information. However, transition modeling adds signal beyond static feature geometry: Markov improves over static Mahalanobis on AGNews, SNIPS, and CLINC150, with the largest gain on SNIPS.

### 4.6.3. SAE ABLATION

We try to extend our method on the raw internal of the transformer with similar setting to our setup, we binarized using the strongest 32 activations to the raw embeddings we utilize the HTM sequential scorer to model the transitions. we report the mean performance for all ag_news splits. Learning on raw internals show bad results, which motivates the use of SAE to disentangle the superposition phenomenon of neurons that might explain such results.

*Table 7.* Mean OOD detection performance across four ID-class runs on the raw internals.

| Method | AUROC ↑ | FPR@95 ↓ |
| --- | --- | --- |
| HTM + raw_activations | 0.753 | 0.638 |

## 5. Discussion, Limitations, and Future Work

We show that *depthwise transition regularities* in sparse, interpretable SAE feature space provide a strong ID-only signal for scope gating, especially in far-OOD and several near-OOD settings. A key limitation emerges when the task taxonomy is substantially finer-grained than the effective resolution of SAE features (e.g., CLINC150 for Gemma2-2B): representational aliasing can yield similar trajectories across distinct intents, degrading discrimination.

This paper focuses on binarized SAE feature. After pooling, SAE activations are density-filtered and Top-$k$ binarized into sparse distributed representations, and the anomaly score is computed from which features are active and how these supports transition across layers. Thus, our detector does not directly use continuous SAE activation magnitudes or SAE reconstruction error as an anomaly signal. A natural next step is to incorporate activation strengths in addition to binary feature supports, which may preserve useful intensity information while retaining the benefits of sparse transition modeling.

Another important direction is to study the role of SAE reconstruction quality more explicitly. Our working hypothesis is that the main benefit of SAEs in this setting comes from inducing a sparse, factorized, and stable discrete support for transition modeling, while reconstruction fidelity is a more indirect upstream property. Comparing SAEs with different reconstruction–sparsity tradeoffs while holding the downstream transition model fixed could help disentangle whether the observed gains arise primarily from reconstruction fidelity, sparsity/factorization, or the induced transition structure.

Finally, our experiments are limited to 16k-dimensional SAEs. Evaluating more expressive dictionaries, such as 65k SAEs, could improve feature resolution and reduce representational aliasing in fine-grained intent settings. However, larger dictionaries may also introduce additional sparsity and coverage tradeoffs, so their effect on transition stability should be evaluated systematically.

**Token pooling and long contexts.** Our current representation mean-pools SAE activations over the token axis and processes at most the first 512 tokens. This deliberately simplifies the detector, but it removes positional structure and may miss highly localized out-of-scope triggers embedded in otherwise in-domain long contexts. Additional long-context experiments suggest that pooled representations do not immediately collapse for longer documents, but they do not establish robustness to arbitrarily long contexts or localized anomalies. Future work should consider span-level, attention-weighted, or hierarchical pooling schemes.

**Cost–interpretability tradeoff.** Our method is not intended to be universally cheaper than output-only OOD scores. Pure output-based methods require only a standard forward pass and are typically less expensive. In contrast, our method requires internal activation extraction and SAE encoding over selected layers. Its advantage is that, once features are extracted, the transition detector is lightweight, requires no base-model fine-tuning, and provides an interpretable white-box signal through decoded SAE feature transitions.

## Impact Statement

This work targets safer and more reliable deployment of LLM-based systems by enabling lightweight *scope gating* that can reject or reroute out-of-domain requests using only in-domain data. Potential risks include uneven false-reject rates across user groups or topics, and false-accepts that allow out-of-scope behavior; both can harm user experience or system safety. Mitigations include careful threshold calibration on representative in-domain traffic, monitoring error patterns over time, and using the gate as one component in a defense-in-depth design (e.g., combined with policy filters and human review for high-stakes actions).

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

## A. Analysis of Layer-wise Domain Cohesion via Top-K Jaccard Similarity

This analysis quantifies the evolution of domain-specific representations across the internal layers of a neural network. By processing the four distinct categories of ag_news dataset —World, Sports, Business, and Sci/Tech—the algorithm identifies how consistently the model activates specific features for samples belonging to the same class. For each layer $l$, the hidden representations are binarized into a set $A$ containing the indices of the $K = 10$ most active neurons, such that $A = \{i \mid h_i \in \text{Top-}K(h)\}$. The similarity between any two samples $i$ and $j$ within a batch is then measured using the Jaccard Index:

$$J(A_i, A_j) = \frac{|A_i \cap A_j|}{|A_i \cup A_j|} \tag{15}$$

To compute this efficiently for a batch of $N$ samples, the intersection is derived from the product of the binarized feature matrix $B$ and its transpose $BB^\top$, while the union is calculated using the principle of inclusion-exclusion: $|A_i \cup A_j| = |A_i| + |A_j| - |A_i \cap A_j|$. The script isolates the off-diagonal elements of the resulting $N \times N$ similarity matrix to avoid self-comparison bias. For every layer, the algorithm aggregates these pairwise scores into a mean $\mu_l$ and a standard deviation $\sigma_l$. These statistics are visualized as a longitudinal plot across three functional stages: the Lexical Zone (Layers 0–4), the Semantic Trunk (Layers 6–18), and the Specific Zone (Layers 20–25). This visualization serves as a diagnostic tool; a rising mean combined with a tightening standard deviation indicates that the model is successfully distilling diverse inputs into a singular, stable semantic concept as the data moves deeper into the architecture.

## B. Top-$K$ Parameter Sweep and Ablation Analysis

This experiment evaluated the impact of feature sparsity (controlled by the Top-$K$ parameter) on Out-of-Distribution (OOD) detection across three architectures: Hierarchical Temporal Memory (HTM), First-Order Markov Chains, and RNNs (LSTM). Evaluations were performed using $16k$ Sparse Autoencoder (SAE) embeddings from `gemma-2-2b`, run on all ag_news splits with density threshold 0.1.

### B.1. Comparative Performance

The results (Table 8) indicate a clear inverse relationship between $K$ and detection accuracy. Peak performance was achieved at the highest sparsity level ($k = 10$) for all methods. Notably, the **First-Order Markov Chain** consistently outperformed or matched more complex architectures, suggesting that local feature transitions are highly discriminative in SAE latent spaces.

*Table 8.* Mean AUROC Performance Averaged Across **ag_news** all ID Classes

| Top-$K$ Value | HTM (Global) | Markov Chain (1st) | RNN (LSTM) |
|---|---|---|---|
| $k = 10$ | 0.9238 | **0.9280** | 0.9131 |
| $k = 32$ | 0.9064 | **0.9195** | 0.9098 |
| $k = 64$ | 0.9002 | **0.9153** | 0.9042 |
| $k = 128$ | 0.5000* | **0.9076** | 0.8788 |

*Note: HTM saturation occurred at $k = 128$, resulting in a total loss of discriminative power.*

### B.2. Key Observations

- **Optimal Sparsity:** $k = 10$ provided the best signal-to-noise ratio. Increasing $K$ likely introduces redundant or "noisy" activations that dilute the class-specific sequence signatures.

- **HTM Saturation:** At $k = 128$ (low sparsity), the HTM model reached 1.0 train/val anomaly means immediately. This indicates the synaptic permanence reached a state where every transition was considered "expected," causing the AUROC to collapse to 0.50.

- **Architecture Robustness:** The Markov Chain exhibited the lowest performance decay as $K$ increased, demonstrating superior robustness compared to the RNN and HTM when dealing with denser activation patterns.

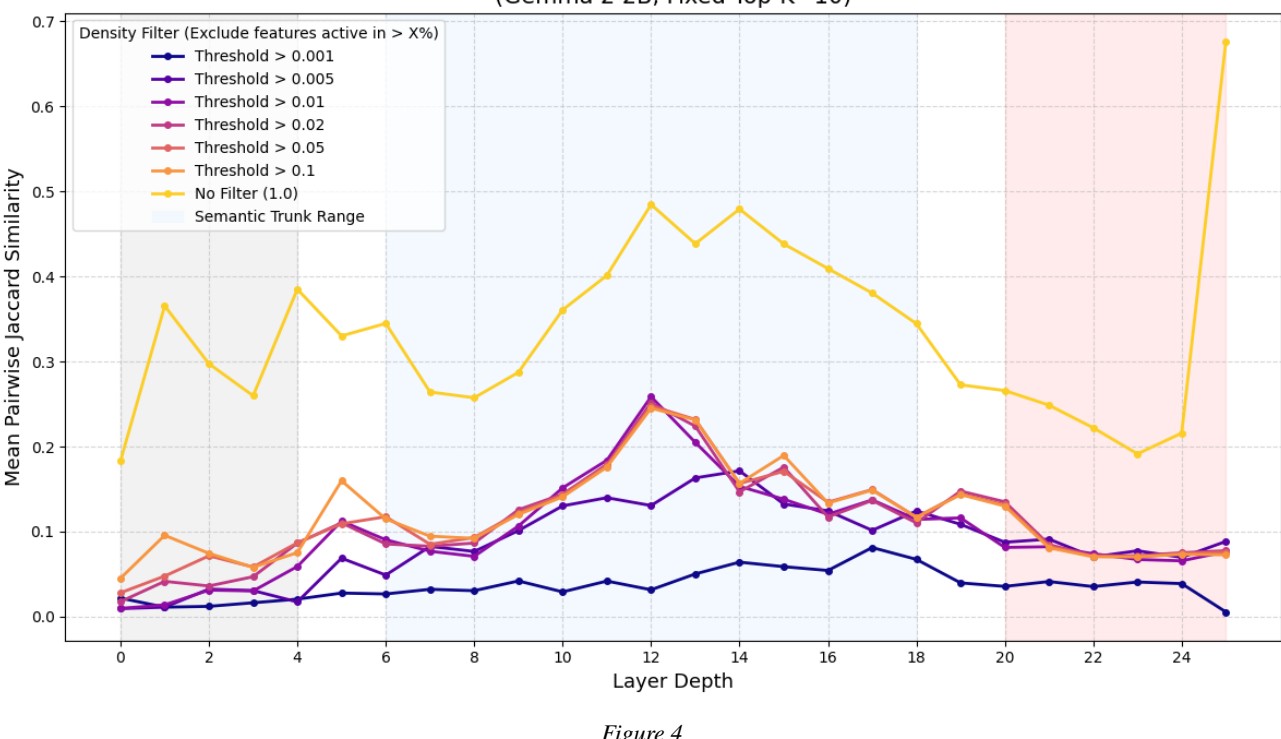

*Figure 4*

## C. Impact of Density Filtering on Semantic features

To investigate the influence of feature sparsity on internal representations, we performed a parameter sweep over density thresholds while maintaining a fixed top-$K$ activation constraint of $K = 10$. Using the Gemma 2 2B model architecture, we computed the Mean Pairwise Jaccard Similarity between binarized sparse distributed representations (SDRs) across all model layers. The density filter serves to exclude ubiquitous features—those active in more than a specified percentage of the PILE dataset—ranging from aggressive filtering (0.1%) to no filtering (100%). Figure 4

## D. Additional Hard-OOD Boundary Examples

We provide additional hard-OOD examples from the AGNews Business/Sci-Tech boundary. These examples further illustrate that low-anomaly OOD inputs often correspond to genuine semantic overlap between the ID and OOD classes.

**Business-ID (2) $\rightarrow$ hard OOD from Sci/Tech (3)**
*"Intel Posts Higher Profit, Sales . . . earnings . . . sales . . . demand . . . "*
**Decoded transition evidence:**

- 18→19: *phrases pertaining to economic trends and conditions $\rightarrow$ references to trade agreements and restrictions*
- 23→24: *phrases related to financial performance and changes in revenue $\rightarrow$ references to organized meetings or discussions related to trade and technology policies*

**Sci/Tech-ID (3) $\rightarrow$ hard OOD from Business (2)**
*"Microsoft Readies Next Business IM Server . . . enterprise instant messaging software . . . "*
**Decoded transition evidence:**

- 16→17: *information related to software updates and their details $\rightarrow$ terms related to electron interactions and superconductivity*
- 16→17: *descriptions and features of new technology products $\rightarrow$ terms related to electron interactions and superconductivity*

# E. Long-Context Stability Analysis

To test whether token-axis pooling collapses on longer documents, we compare pooled SAE representations from the first 512 tokens with longer prefixes of the same 20NG examples. We report Spearman correlation and cosine similarity between the resulting pooled representations.

*Table 9.* Stability of pooled SAE representations as input length increases.

| Comparison | Spearman $\rho$ | Cosine similarity |
|---|---|---|
| 512 vs. 1024 tokens | 0.874 | 0.933 |
| 512 vs. 2048 tokens | 0.769 | 0.860 |

The representation remains strongly aligned from 512 to 1024 tokens, with more noticeable degradation at 2048 tokens. This suggests that truncation and pooling do not immediately destroy document-level semantic information, but the degradation at longer prefixes motivates more structured pooling mechanisms in future work.

*Table 10.* Long-only versus short-matched 20NG experiment. Results report AUROC. The long-only subset contains documents whose original length exceeds 512 tokens; the short-matched control matches the long-only setting in sample count and label distribution.

| OOD dataset | Short-matched | Long-only |
|---|---|---|
| CLINC150 | $0.749 \pm 0.009$ | 0.918 |
| SST-2 | $0.904 \pm 0.006$ | 0.929 |
| MNLI | $0.886 \pm 0.007$ | 0.899 |
| IMDB | $0.909 \pm 0.010$ | 0.808 |
| RTE | $0.940 \pm 0.003$ | 0.835 |

The mixed pattern indicates that the method does not simply fail on long documents, but it also does not guarantee robustness to all long-context settings. Performance improves for some OOD sources and degrades for others, suggesting that long-context behavior depends on the semantic composition of the document and the OOD source.

