# OpenReview forum: "Domain Restriction via SAE Multi-Layer Transitions"
_ICML.cc/2026/Conference — ICML 2026 regular_

### Official Review · Reviewer_zQ5J · 2026-03-12

**Soundness:** 3
**Presentation:** 2
**Significance:** 2
**Originality:** 2
**Overall Recommendation:** 4
**Confidence:** 4

**Summary:**

The paper proposes a lightweight and training free method for out of distribution detection in large language models by tracking the layer by layer transition probabilities of internal features extracted via sparse autoencoders.

**Compliance With Llm Reviewing Policy:**

Affirmed.

**Final Justification:**

The authors did a good job addressing my questions by providing extra experiments on long contexts and clarifying that they actually only use a subset of deeper layers instead of all of them. They honestly acknowledged the limitations of mean pooling and promised to add the runtime details, so my main concerns are resolved.

**Key Questions For Authors:**

- How does the system perform when the input sequence length scales up to thousands of tokens, given the current pooling mechanism?
- Could you provide an analysis of the computational overhead introduced by running SAEs across all layers during inference?

**Limitations:**

yes

**Strengths And Weaknesses:**

## Strengths
- Uses internal SAE features for interpretability, avoiding black box confidence scores.
- Achieves competitive performance using only in distribution data without requiring fine tuning.
- The inclusion of a detailed case study effectively highlights how the method captures true semantic overlap.

## Weaknesses
- My main concerns lie in the token axis mean pooling strategy, which collapses the entire sequence into a single vector per layer and fundamentally destroys spatial and structural information. When dealing with complex or long context user prompts, this compression might dilute localized triggers within a broader background, causing the Markov transition model to miss crucial semantic anomalies.
- The method struggles with fine grained intents like CLINC150, and merely scaling SAE dictionary size might lead to extreme sparsity issues.
-  The reliance on training data statistics might confuse benign style shifts with true semantic out of bounds behavior.

---

> ### Author Rebuttal · Authors · 2026-03-30
>
> We thank the reviewer for the thoughtful feedback and for the positive assessment of the paper’s interpretability, training-free setup, and case-study analysis. We agree that long-context behavior, token-axis mean pooling, and inference-time cost are important questions, and we will clarify both these points and the method’s current limitations in the revision.
>
> ### 1. Long inputs and the effect of mean pooling
>
> We agree that mean pooling over the token axis removes positional structure and may be less sensitive to highly localized triggers within long contexts. Our claim is therefore limited: under the current setup, pooling does not appear to fail immediately once documents become longer than 512 tokens, but we do not claim robustness to arbitrarily long contexts or sharply localized anomalies.
>
> Our current system processes at most the first 512 tokens per example. To test whether this truncation causes representation collapse on long inputs, we performed a token-growth analysis on long 20NG documents by comparing the pooled SAE representation from the first 512 tokens against longer prefixes of the same example:
>
> | Comparison | Spearman rho | Cosine similarity |
> |---|---:|---:|
> | 512 vs. 1024 tokens | 0.874 | 0.933 |
> | 512 vs. 2048 tokens | 0.769 | 0.860 |
>
> These results suggest that the pooled representation remains well aligned from 512 to 1024 tokens, with more noticeable degradation only at much longer comparisons such as 2048 tokens. Thus, the current evidence does not support a collapse story in which truncation alone breaks the method.
>
> We also ran a complementary long-only experiment on 20NG. We trained the method on samples whose original documents exceeded 512 tokens, and compared this against a short-text control set with matched label distribution (mean across 5 seeds). The pattern is mixed: the long-only setup performs better on CLINC150 and SST-2, is roughly comparable on MNLI, and is worse on IMDB and RTE.
>
> | OOD dataset | Short-matched | Long-only |
> |---|---:|---:|
> | CLINC150 | 0.749±0.009 | 0.918 |
> | SST-2 | 0.904±0.006 | 0.929 |
> | MNLI | 0.886±0.007 | 0.899 |
> | IMDB | 0.909±0.010 | 0.808 |
> | RTE | 0.940±0.003 | 0.835 |
>
> This again suggests that the method does not simply break on long examples, but it also does not establish robustness to very long contexts or to highly localized semantic cues. We will make this limitation explicit in the revised paper.
>
> Finally, we qualitatively inspected the highest-activating SAE features for long prompts and found that they remain semantically coherent across layers. For example, in a long 20NG sample:
>
> _"Size of armies, duration, numbers of casualties both absolute and as a percentage of those involved, geographical area..."_
>
> highly activated features consistently referred to themes such as **military actions**, **war and its consequences**, and **historical conflicts in Europe**, and **geopolitical tensions**. This gives further evidence that the deeper-layer pooled representation still captures document-level semantics even for long inputs.
>
> ### 2. Computational overhead
>
> We also agree that computational overhead is important. In our setup, we do **not** run SAEs over all layers, but only over a contiguous block of deeper layers (Gemma-2-2B/9B, 16k SAE, layers 16--24/38--40), using up to the first 512 tokens. Empirically, the dominant cost is feature extraction, while transition fitting and scoring are comparatively small. We will report the runtime breakdown explicitly in the revision so that this overhead is clear.
>
> ### 3. Fine-grained intent distinctions and style shifts
>
> We agree with the reviewer that fine-grained intent distinctions, such as those in CLINC150, remain challenging. We also agree that benign stylistic shifts may sometimes be conflated with semantic out-of-bounds behavior. Our use of deeper layers is intended to reduce sensitivity to purely surface variation, but it does not eliminate this issue. We will revise the paper to make these limitations clearer and to avoid overstating the method’s robustness in settings with subtle intent boundaries or highly localized trigger patterns.
>
> ### 4. Planned revision
>
> In the revision, we will:
>
> - clarify that the method uses a subset of deeper layers, not all layers;
> - add the long-context stability analysis and the long-only vs. short-matched comparison;
> - report the runtime breakdown explicitly;
> - state more clearly that mean pooling is a deliberate simplification with tradeoffs, especially for highly localized anomalies in long contexts.
>
> We thank the reviewer again for the helpful feedback. We believe these additions will make the scope, strengths, and limitations of the method much clearer.

---

> > ### Author Rebuttal · Reviewer_zQ5J · 2026-04-03
> >
> > The authors did a good job addressing my questions by providing extra experiments on long contexts and clarifying that they actually only use a subset of deeper layers instead of all of them. They honestly acknowledged the limitations of mean pooling and promised to add the runtime details, so my main concerns are resolved.

---

> > > ### Author Response · Authors · 2026-04-03
> > >
> > > Thank you very much for your positive acknowledgment.
> > > We are very happy that it fully answers your concerns.
> > > As a result, will you consider adjusting your score to reflect it?
> > > Thanks in advance.

---

### Official Review · Reviewer_2qdx · 2026-03-12

**Soundness:** 3
**Presentation:** 3
**Significance:** 3
**Originality:** 3
**Overall Recommendation:** 4
**Confidence:** 3

**Summary:**

This paper introduces a white-box approach to detect OOD interactions in LLMs The authors use Sparse Autoencoders (SAEs) to track internal layer transitions and extract domain-specific signatures. This lightweight method effectively identifies OOD inputs, improves the interpretability of the model's internal decision-making, and is validated on Gemma-2 (2B and 9B) models.

**Compliance With Llm Reviewing Policy:**

Affirmed.

**Key Questions For Authors:**

see weaknesses

**Limitations:**

yes

**Strengths And Weaknesses:**

Strengths:

* Well-motivated: Out-of-domain interactions with LLMs are a significant issue.

* Novel in my view: Instead of inferring from input/output data, the authors track internal layer transitions to monitor OOD interactions with LLMs.

Weaknesses:

* No major concerns.
* I would like to ask about the computational cost and success rate of traditional methods relying on input/output text compared to this SAE-based detection method. I think this method is novel, but I am unsure about its practicality compared to existing methods.

---

> ### Author Rebuttal · Authors · 2026-03-30
>
> We thank the reviewer for the positive assessment and for raising the important question of practicality relative to traditional input/output-based OOD detectors.
>
> We agree that computational cost is an important consideration. A first clarification is that our method does **not** run SAEs over all transformer layers. In the main setting, we extract SAE features only from a contiguous block of deeper layers (Gemma-2-2B, 16k SAE, layers 16--24), using up to the first 512 tokens.
>
> To make the practical overhead explicit, we now report an end-to-end benchmark on a representative setup: 100 ID training samples, 50 ID test samples, and 50 OOD test samples, all with at least 512 raw tokens, using layers [16,24] and a 16k SAE.
>
> | Stage | Time (s) |
> |---|---:|
> | Train extraction | 57.43 |
> | Train fit | 1.58 |
> | ID extraction | 29.27 |
> | ID scoring | 0.0945 |
> | OOD extraction | 25.45 |
> | OOD scoring + metrics | 0.0990 |
> | Full pipeline total | 151.92 |
>
> The main takeaway is that **feature extraction dominates runtime**, while transition-model fitting and scoring are comparatively negligible. In other words, once SAE activations have been extracted, the Markov detector itself adds very little overhead.
>
> We also evaluated a more realistic long-versus-short setting on 20NG using the same pipeline. In the long-only subset, we retain the **1129** out of **11314** training documents whose original length exceeds **512 tokens**. In the short-matched control, we sample **1129** documents from the remaining **10185** documents with length at most **512 tokens**, matching the long-only condition in per-label counts. Train extraction took **346.93 s** in the short-matched control and **725.54 s** in the long-only subset, while fitting remained small in both cases (**4.25 s** vs. **4.46 s**). This shows that increasing input length substantially increases runtime, and that this increase is driven almost entirely by representation extraction rather than by the detector itself.
>
> Regarding comparison to traditional methods based only on input/output text, we deliberately focused on the LR baseline from Zhang et al. (2025) because it is the strongest prior baseline reported in this line of work, and that paper already compares against a broader set of classical confidence-based and training-based methods. We chose this comparison to keep the paper focused while still benchmarking against a strong reference point.
>
> More broadly, these approaches occupy different points in the cost/interpretability tradeoff space. Pure output-based methods are typically the cheapest, since they require only a standard forward pass and a score computed from the output. Methods such as TAPT or supervised contrastive learning can have substantially higher offline cost because they require additional adaptation and training. Our method lies between these extremes: it is more expensive than pure output-based scoring because it requires internal SAE feature extraction, but the detector itself is lightweight, training-free, and inexpensive once features are extracted.
>
> We therefore do not claim that our approach is universally cheaper than all input/output-based alternatives. Rather, we claim that it offers a practical tradeoff: the main overhead is extracting internal features from a limited set of deeper layers, while the detector itself is simple, fast, and provides an interpretable white-box signal that text-only detectors do not.
>
> In terms of effectiveness, our paper already compares against the strongest prior reported baseline in this line of work, rather than only against weaker classical methods. We will make this motivation clearer in the revision, and we will add the runtime breakdown above so that the practical overhead of our method is explicit.
>
> We thank the reviewer again for the helpful feedback and practical question. We will incorporate these clarifications and the runtime breakdown in the revision.

---

### Official Review · Reviewer_g4kc · 2026-03-13

**Soundness:** 2
**Presentation:** 3
**Significance:** 2
**Originality:** 3
**Overall Recommendation:** 4
**Confidence:** 4

**Summary:**

This paper addresses domain restriction for LLM-based agents: deciding whether a user request is in scope for a domain-specific agent or should be rejected or rerouted. The authors propose modeling transitions of sparse SAE features across transformer layers, using residual activations from Gemma-2 models mapped into SAE feature space and scored mainly with a first-order Markov model trained only on in-domain data. Experiments on far-OOD and near-OOD benchmarks show that layer-transition modeling improves over static feature-based variants and that SAE-based representations work better than raw activations, suggesting that sparse layerwise transitions can provide a useful signal for scope detection.

**Compliance With Llm Reviewing Policy:**

Affirmed.

**Final Justification:**

Thanks the authors for providing additional details and results. I think the discussion of why SAE helps beyond interpretability is valuable. However, this paper combines multiple techniques, and I believe a more rigorous investigation of why certain techniques are critical would strengthen the work. For example, in the explanation paragraph of the “Why SAE Helps” section, one additional question is what role SAE reconstruction error plays.

Given the above, I am inclined to keep my original score.

**Key Questions For Authors:**

How does the method compare with simpler non-SAE baselines, or with just using a strong multi-layer sparse representation without transition modeling? This would help clarify whether the gain comes from the Markov transition modeling itself, or simply from using a good intermediate representation.

**Limitations:**

yes

**Strengths And Weaknesses:**

## Strengths
- This paper addresses domain restriction for LLM-based agents by proposing to model transitions of sparse SAE features across transformer layers with a first-order Markov model trained only on in-domain data. This is a novel and interesting angle, since it uses internal layerwise dynamics rather than standard black-box confidence or final-layer representations for scope detection.
- Various ablations and evaluations are performed to support the method. The paper includes far-OOD and near-OOD evaluations, comparisons across model sizes, ablations over Markov, HTM, and RNN variants, and comparisons between SAE-based features and raw activations. These experiments help support the claim that transition structure in SAE feature space contains useful information for domain restriction.

## Weaknesses
- The evaluation gain from this method is not very significant. While the method shows improvement over some internal variants, the advantage over stronger baselines is limited, and in several settings the results are only competitive rather than clearly better. This weakens the case for the practical value of the method.
- No baseline is included using simpler methods, such as linear probing, or other non-SAE methods, such as directly using dense activations. Because of this, it is hard to tell whether the benefit comes specifically from the proposed transition modeling or more generally from using a reasonable intermediate representation.
- The discussion of the effectiveness of using SAE is only high level. The paper provides intuition that SAE features are sparse, aligned, and more interpretable, but does not offer a more fundamental explanation of why SAE is especially helpful in this setting or what property of SAE is driving the improvement.

---

> ### Author Rebuttal · Authors · 2026-03-30
>
> We thank the reviewer for the thoughtful and constructive feedback. We appreciate the recognition that modeling layerwise SAE-feature transitions is a novel white-box approach, and we agree that the key questions are (i) whether the gains come from transition modeling rather than simply from using a strong intermediate representation, and (ii) why SAE is especially helpful in this setting.
>
> ### 1. Practical value of the gains
>
> We agree that the method is not uniformly stronger than the strongest prior baselines, and we will revise the paper to make this clearer. At the same time, our method is **training-free**, whereas the LR baseline requires an additional fitting stage. Viewed in that light, we believe the results are encouraging, and suggest that pretrained LLM representations already contain substantial information relevant to domain restriction.
>
> On near-OOD AGNews, our Markov method achieves an average AUROC of **0.915** across the four in-domain classes, compared with **0.787** for LR. On SNIPS, our method reaches **0.960** versus **0.975** for LR, i.e., competitive performance without gradient updates. We agree that **CLINC150** remains challenging (**0.830** for our method vs. **0.866** for LR), and we will state this limitation more explicitly in the revision. We also agree that far-OOD is less informative here, since several methods are already near ceiling.
>
> ### 2. Does the gain come from transition modeling, or simply from a good representation?
>
> This is a fair and important question. To address it more directly, we added a simpler baseline under the **same representation setup** (Gemma-2-2B, 16k SAE, layers 16--24):
>
> - **Static Mahalanobis:** pooled SAE activations across layers, but without transition modeling.
>
> Concretely, this baseline uses the same SAE representation as our method, but removes the transition component: we fit a **Gaussian** to the training distribution at each layer, compute a Mahalanobis score at each layer, and average the scores across layers. This comparison is designed to isolate whether multi-layer SAE features alone are sufficient.
>
> | Method | AGNews avg. AUROC | SNIPS | CLINC150 |
> |---|---:|---:|---:|
> | LR baseline (supervised) | 0.787 | **0.975** | **0.866** |
> | Static Mahalanobis (SAE, no transitions) | 0.905 | 0.862 | 0.791 |
> | Markov (ours) | **0.915** | 0.960 | 0.830 |
>
> We believe this comparison clarifies the source of the gain.
>
> - **Multi-layer SAE features alone are strong, but not sufficient.** The static Mahalanobis baseline is already competitive on AGNews, showing that the SAE representation itself is meaningful.
> - **Transition modeling adds signal beyond static pooling.** On SNIPS, Markov improves AUROC from **0.862** to **0.960** over the static SAE baseline; on CLINC150, it improves from **0.791** to **0.830**; and on AGNews, it improves average AUROC from **0.905** to **0.915**.
>
> Overall, the evidence suggests that the improvement is **not** only due to using a reasonable intermediate representation. Rather, the SAE representation is helpful, and explicit modeling of how features transition across layers adds discriminative information on top of that representation.
>
> ### 3. Why SAE helps beyond interpretability
>
> We agree that the original discussion was too high-level, and we will make the mechanism more explicit.
>
> A key result already in the paper is the raw-activation ablation: keeping the same transition pipeline but replacing SAE features with raw LLM activations causes a substantial drop on AGNews, from **0.912** to **0.753** mean AUROC, while FPR95 worsens from **0.384** to **0.638**. This suggests that SAE is important to performance, not only to interpretability.
>
> Our current interpretation is that SAE helps because it maps the dense residual stream into a **sparse, more factorized feature space**. In raw activation space, activity is diffuse and entangled, making layer-to-layer "feature transitions" difficult to characterize reliably. In SAE space, only a small subset of features is active for a given input, yielding a more discrete set of activation events. This makes transition statistics more stable and informative, which is precisely the regime in which a Markov-style model becomes useful.
>
> ### 4. What we will add in the revision
>
> To make this clearer in the final paper, we will:
>
> 1. add the new **static Mahalanobis** baseline;
> 2. clarify that the main advantage of the method is that it is **training-free**;
> 3. expand the discussion of **why SAE helps**, emphasizing sparsity/factorization rather than interpretability alone;
> 4. state more clearly that the gains are strongest in some near-OOD settings and more limited in others.
>
> We thank the reviewer again for the helpful feedback. We believe these additions make the contribution clearer: SAE features provide a sparse representation well suited to transition modeling, and explicit layer-transition modeling contributes signal beyond static multi-layer representations alone.

---

> > ### Author Rebuttal · Reviewer_g4kc · 2026-04-04
> >
> > Thanks the authors for providing additional details and results. I think the discussion of why SAE helps beyond interpretability is valuable. However, this paper combines multiple techniques, and I believe a more rigorous investigation of why certain techniques are critical would strengthen the work. For example, in the explanation paragraph of the “Why SAE Helps” section, one additional question is what role SAE reconstruction error plays.
> >
> > Given the above, I am inclined to keep my original score.

---

> > > ### Author Response · Authors · 2026-04-06
> > >
> > > Thank you for the follow-up and for clarifying the point about reconstruction error. In our rebuttal, we understood the main concern to be whether the gains came from transition modeling itself versus simply using a strong intermediate representation, and we therefore added the static SAE baseline to isolate that question more directly.
> > >
> > > Regarding SAE reconstruction error, this is a valuable follow-up question, but it was not the mechanism we focused on in the current study because our downstream detector does not use continuous SAE activation strengths directly. In our pipeline, pooled SAE activations are density-filtered and then top-k binarized into a sparse distributed representation, and the anomaly score is computed from which features are active and how they transition across layers, rather than from their exact magnitudes. For that reason, our working hypothesis has been that the main benefit of SAE here comes from producing a sparse, factorized, and stable discrete support for transition modeling, while reconstruction fidelity is a more indirect upstream property.
> > >
> > > We agree, however, that studying the role of reconstruction error more explicitly would be valuable future work. In particular, comparing SAEs with different reconstruction quality while holding the downstream transition model fixed could help further disentangle whether the benefit comes primarily from reconstruction fidelity, sparsity/factorization, or the induced transition structure.

---

### Decision · Program_Chairs · 2026-04-30

**Decision:**

Accept (regular)

**Comment:**

The reviews for this paper were consistently positive, with all reviewers placing it in the weak-accept range. Reviewers found the problem well motivated and the technical approach novel: instead of relying on black-box confidence signals, the paper models transitions of SAE-derived internal features across layers to detect whether a query is in or out of domain. The method was viewed as lightweight, interpretable, and competitive, with a particularly appealing property that it can be trained using only in-domain data.
The main concerns were not about correctness, but about analysis depth and completeness. In particular, reviewers asked for a clearer disentangling of which components are responsible for the gains, stronger explanation of why SAE features help beyond interpretability, and more clarity on pooling, runtime, and evaluation details. The rebuttal addressed most of these points with additional experiments and clarifications, and one reviewer explicitly stated that their main concerns were resolved. One reviewer still wanted a more rigorous causal explanation of the ingredients, but this reads as a limitation of completeness rather than a fatal weakness. Overall, I find the paper technically solid, novel enough, and likely to be useful to researchers working on controllable or domain-specific LLM systems.